# A Framework for the Quantitative Evaluation of Disentangled Representations

**Cian Eastwood**
School of Informatics
University of Edinburgh, UK
c.eastwood@ed.ac.uk

**Christopher K. I. Williams**
School of Informatics
University of Edinburgh, UK
and Alan Turing Institute, London, UK
ckiw@inf.ed.ac.uk

## Abstract

Recent AI research has emphasised the importance of learning *disentangled* representations of the explanatory factors behind data. Despite the growing interest in models which can learn such representations, visual inspection remains the standard evaluation metric. While various desiderata have been implied in recent definitions, it is currently unclear what exactly makes one disentangled representation better than another. In this work we propose a framework for the quantitative evaluation of disentangled representations when the ground-truth latent structure is available. Three criteria are *explicitly* defined and *quantified* to elucidate the quality of learnt representations and thus compare models on an equal basis. To illustrate the appropriateness of the framework, we employ it to compare quantitatively the representations learned by recent state-of-the-art models.

## 1 Introduction

To gain a conceptual understanding of our world, models must first learn to understand the factorial structure of low-level sensory input without supervision (Bengio et al., 2013; Lake et al., 2016; Higgins et al., 2017). As argued in several notable works (Desjardins et al., 2012; Bengio et al., 2013; Chen et al., 2016; Higgins et al., 2017), this understanding can only be gained if the model learns to *disentangle* the underlying explanatory factors hidden in unlabelled input.

A disentangled representation is generally described as one which separates the factors of variation, explicitly representing the important attributes of the data (Desjardins et al., 2012; Bengio et al., 2013; Cohen & Welling, 2014b; Kulkarni et al., 2015; Chen et al., 2016; Higgins et al., 2017). For example, given an image dataset of human faces, a disentangled representation may consist of separate dimensions (or features) for the face size, hairstyle, eye colour, facial expression, etc. Ultimately, we would like to learn representations that are invariant to irrelevant changes in the data. However, the relevant downstream tasks are generally unknown at training time and hence it is difficult to deduce a priori which features will be useful. Thus, the most robust method is to disentangle as many factors of variation as possible, discarding as little information as possible (Desjardins et al., 2012; Bengio et al., 2013).

Despite the expanding literature on models which seek to learn disentangled representations (Desjardins et al., 2012; Reed et al., 2014; Zhu et al., 2014; Cheung et al., 2014; Larsen et al., 2015; Makhzani et al., 2015; Yang et al., 2015; Kulkarni et al., 2015; Whitney et al., 2016; Chen et al., 2016; Higgins et al., 2017; Denton & Birodkar, 2017), visual inspection remains the standard evaluation metric. While the work of Higgins et al. (2017) partially addresses this issue (as discussed in section 3) and various definitions have implied additional desiderata like interpretability (Bengio et al., 2013; Kulkarni et al., 2015; Chen et al., 2016), invariance (Goodfellow et al., 2009; Cohen & Welling, 2014a;b; Lenc & Vedaldi, 2015) and equivariance (Kivinen & Williams, 2011; Lenc & Vedaldi, 2015; Jayaraman & Grauman, 2015), current research generally lacks a clear metric for quantitatively evaluating and comparing disentangled representations.

In this work we propose a framework for the quantitative evaluation of disentangled representations when the ground-truth latent structure is available. To elucidate the quality of learnt representations

and thus compare models on an equal basis, desiderata of disentangled representations are *explicitly* defined and *quantified*. These unified desiderata help define the disentangled representations which we seek and remove the need for a subjective visual evaluation by a human arbiter. To illustrate the appropriateness of this framework, we employ it to compare quantitatively the representations learned by principal components analysis (PCA), the variational autoencoder (VAE, Kingma & Welling 2013), $\beta$-VAE (Higgins et al., 2017) and information maximising generative adversarial networks (InfoGAN, Chen et al. 2016).

In the remainder of this paper, we begin by detailing the theoretical framework and how it facilitates the quantitative evaluation of disentangled representations. Next we review related desiderata and metrics for evaluating disentangled representations. Finally, we describe the dataset and model specifics before presenting the experimental results.

## 2  THEORETICAL FRAMEWORK

Models for disentangled factor learning seek a compact data representation or *code* $c$ of dimension $D$, which consists of disentangled and interpretable latent variables. For synthetic data, the $K$-dimensional generative factors $z$ are designed to be an ideal such representation. Thus if $D = K$ the ideal disentangled code $c^*$ should be some (scaled) permutation of $z$, i.e. they should be related by a generalised permutation matrix (or monomial matrix[1]). If $D > K$, one would expect to obtain this monomial structure along with a number of 'dead' or irrelevant units in $c$ which are not predictive of / informative about $z$. Thus, we can quantitatively evaluate the codes learned by a given model $M$ using the following steps:

1. Train $M$ on a synthetic dataset with generative factors $z$
2. Retrieve $c$ for each sample $x$ in the dataset ($c = M(x)$)
3. Train regressor $f$ to predict $z$ given $c$ ($\hat{z} = f(c)$)
4. Quantify $f$'s deviation from the ideal mapping and the prediction error

We now detail the proposed evaluation metrics, i.e., steps 3 and 4. We train $K$ regressors to predict the value of $K$ generative factors. The regressor $f_j$ predicts $z_j$ given $c$, that is, it learns a mapping $f_j(c) : \mathbb{R}^D \to \mathbb{R}^1$. We use regressors that can provide a matrix of relative importances $R$, where $R_{ij}$ denotes the relative importance of $c_i$ in predicting $z_j$ (see section 4.3). This allows us to explicitly define and quantify three criteria of disentangled representations or *codes* which are implicit in recent definitions (Desjardins et al., 2012; Bengio et al., 2013; Kulkarni et al., 2015; Chen et al., 2016; Higgins et al., 2017), namely *disentanglement*, *completeness* and *informativeness*.

**Disentanglement.**  The degree to which a representation factorises or *disentangles* the underlying factors of variation, with each variable (or dimension) capturing at most one generative factor. The disentanglement score $D_i$ of code variable $c_i$ is quantified by $D_i = (1 - H_K(P_{i.}))$, where $H_K(P_{i.}) = -\sum_{k=0}^{K-1} P_{ik} \log_K P_{ik}$ denotes the entropy and $P_{ij} = R_{ij} / \sum_{k=0}^{K-1} R_{ik}$ denotes the 'probability' of $c_i$ being important for predicting $z_j$. If $c_i$ is important for predicting a single generative factor, the score will be 1. If $c_i$ is equally important for predicting all generative factors, the score will be 0. $D_i$ can be visualised by examining row $i$ of the Hinton diagrams as in Figure 3.

In order to account for dead or irrelevant units in $c$, relative code variable importance $\rho_i = \sum_j R_{ij} / \sum_{ij} R_{ij}$ is used to construct a *weighted* average $\sum_i \rho_i D_i$ expressing overall disentanglement. If a code variable $c_i$ is irrelevant for predicting $z$, then its $\rho_i$ (and thus contribution to the overall disentanglement) will be near zero.

**Completeness.**  The degree to which each underlying factor is captured by a single code variable. The completeness score $C_j$ in capturing generative factor $z_j$ is quantified by $C_j = (1 - H_D(\tilde{P}_{.j}))$, where $H_D(\tilde{P}_{.j}) = -\sum_{d=0}^{D-1} \tilde{P}_{dj} \log_D \tilde{P}_{ij}$ denotes the entropy of the $\tilde{P}_{.j}$ distribution. If a single code variable contributes to $z_j$'s prediction, the score will be 1 (complete). If all code variables equally

---

[1] A matrix is monomial if there is exactly one non-zero element in each row and column. If the non-zero elements have value 1 the matrix is a permutation matrix.

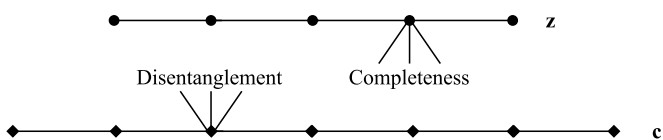

Figure 1: **Visualising disentanglement and completeness.**

contribute to $z_j$'s prediction, the score will be 0 (maximally overcomplete). $C_j$ can be visualised by examining column $j$ of the Hinton diagrams as in Figure 3.

**Informativeness.** The amount of information that a representation captures about the underlying factors of variation. To be useful for natural tasks which require knowledge of the important attributes of the data (e.g. object recognition), representations must ultimately capture information about the underlying factors of variation (Bengio et al., 2013; Chen et al., 2016). The informativeness of code $c$ about generative factor $z_j$ is quantified by the prediction error $E(z_j, \hat{z}_j)$ (averaged over the dataset), where $E$ is an appropriate error function and $\hat{z}_j = f_j(c)$. It is important to note that the prediction error $E(z_j, \hat{z}_j)$, and thus this informativeness metric, is dependent on the capacity of $f$, with linear regressors only capable of extracting information about $z$ in $c$ that is explicitly represented. Hence this informativeness metric is also dependent on a model's ability to explicitly represent information about $z$ in $c$, which in turn is dependent on the model's ability to disentangle the underlying factors of variation ($z$). Thus the informativeness metric has some overlap with the disentanglement metric, with the size of the overlap determined by the capacity of $f$ (no overlap with infinite capacity).

While the disentanglement score quantifies the number of generative factors captured by a given code variable, the completeness score quantifies the number of code variables which capture a given generative factor. Together, these scores quantify the deviation from the ideal one-to-one mapping between $z$ and $K$ of the dimensions in $c$. Figure 1 illustrates this idea.

Despite the overlap between the disentanglement and informativeness metrics with low-capacity linear regressors, these are ultimately *distinct* criteria. While disentanglement requires each code variable in $c$ to be only perturbed by changes in a single $z$, informativeness requires these perturbations to be systematic and thus *informative*. This motivates the use of non-linear regressors in section 4.3.

While the ideal code would be able to explicitly represent each generative factor with a single variable, models with generic priors cannot be expected to learn such complete and explicit codes. For example, generative factors which are drawn from a distribution on a circle cannot be accurately captured by single code variables on which unwrapped prior distributions are imposed. Thus, with generic priors like the standard normal, information about such topologically distinct generative factors may be non-linearly encoded across multiple code variables. Empirical results in Appendix C and (Higgins et al., 2017, fig. 7) support this idea, with several code variables resembling non-linear functions (like the sine and cosine) of the object azimuth. This further motivates the use of non-linear regressors in section 4.3.

Our criteria assume that it is possible to recover the latent factors $z$ from the data. If the data $x$ depends on a *linear* combination of (some of) the underlying $z$'s with a spherically symmetric distribution, then it will only be possible to recover these components up to a rotation matrix. This is the well-known issue of the rotation of factors in the linear factor analysis model (see e.g., Mardia, Kent, and Bibby 1979, sec. 9.6), and also leads to the condition in independent components analysis (ICA) that at most one of the $z$'s can be Gaussian (Hyvärinen et al., 2001). In this case, the informativeness metric remains valid but the disentanglement and completeness metrics do not as they are dependent on the arbitrary rotation which determines $c$'s alignment with $z$. Although $z$ may be used to compute the rotation matrix which best aligns $c$ and $z$, we ultimately wish to evaluate models which will not have access to $z$ at test time.

## 3   RELATED WORK

The question of how well a learned representation $c$ matches the true generative factors $z$ has been considered in the 'square' case of independent components analysis (ICA), where $D = K$. In the ICA case, the data is generated as $x = Az$ and the learned representation is obtained as $c = Wx$, where $A$ is the mixing matrix and $W$ is the learned 'un-mixing' matrix. Ideally $P = WA$ will be equal to a permutation matrix. Yang & Amari (1997, sec. 6.1) propose an error metric to assess how close $P$ is to a permutation matrix[2]. This metric takes the form

$$E = \sum_i \left( \sum_j \frac{|p_{ij}|}{\max_k |p_{ik}|} - 1 \right) + \sum_j \left( \sum_i \frac{|p_{ij}|}{\max_k |p_{kj}|} - 1 \right), \tag{1}$$

summing two terms which have similar goals to our disentanglement and completeness scores respectively, although expressed by comparing with the maximum value in the row or column, rather than via an entropic measure. Note that, due to the linear structure of ICA, there is no explicit mapping between $c$ and $z$. We report separate scores as they capture distinct criteria and go beyond this metric by handling the non-square case when $D > K$.

Predicting $z$ from $c$ has also been considered previously. Higgins et al. (2017, sec. A.5) use a linear classifier to predict discrete settings of $z$ and thus quantify the amount of explicit information about $z$ in $c$, albeit with a discretisation step which we find unnecessary. Higgins et al. (2017, sec. 3) also propose a disentanglement metric. With this method, one of the generative factors say $z_k$ is held fixed, and pairs of $x$'s are drawn, generated with different random $z$'s except for the fixed $z_k$. Pairwise absolute differences of the resulting codes $|c_1 - c_2|$ are then computed and averaged over repetitions before being used to train a linear classifier to predict which generative factor was held fixed. In our view this is unnecessarily cumbersome—by setting up a regression problem to predict $z$ from $c$ as we have done, the structure of the $R_{ij}$ matrix can be interrogated to quantify the degree of disentanglement. In addition, this facilitates the quantification of additional criteria, namely completeness and informativeness, without needing to generate any additional datasets.

Glorot et al. (2011, fig. 3) predict $z$ from $c$ using a lasso regressor but only to qualitatively assess disentanglement, visually assessing the overlap of important features for the separate tasks of domain recognition and sentiment classification. Karaletsos et al. (2015) do so with an unspecified regressor, thus quantifying informativeness. In addition, they devise a quantitative metric to determine a model's ability to disentangle the underlying factors of variation in images. In particular, they predict the order of query-specific oracle triplets of images, where the order indicates image similarity with respect to a query (e.g., 'Where is the light condition most similar in terms of azimuth?'). However, the proposed metric is specifically designed to evaluate their 'oracle-prioritized belief network' and thus overly cumbersome to be used as a generic disentanglement metric.

Properties such as invariance and equivariance have been proposed as desiderata for representations or codes (Goodfellow et al., 2009; Kivinen & Williams, 2011; Cohen & Welling, 2014b; Lenc & Vedaldi, 2015; Jayaraman & Grauman, 2015). In our view these qualities arise naturally from a properly disentangled and informative code. Consider, for example, the code of an object which consists of separate variables for its class (e.g., cup, bottle, banana etc.), position, pose, texture etc. If the object is translated, its position code variable(s) will transform accordingly (equivariance), but other code variables will remain invariant.

## 4   EXPERIMENTS

We employ the framework to compare quantitatively the codes learned by PCA, the VAE, $\beta$-VAE and InfoGAN. The results can be reproduced with our open source implementation[3].

### 4.1   DATA

We use the graphics renderer described in (Moreno et al., 2016) to generate 200,000 $64 \times 64$ colour images of an object (teapot) with varying pose and colour (see Figure 2). For simplicity, the camera

---

[2]We thank Andriy Mnih for pointing out to us the work of Yang and Amari.
[3]Code and dataset available at `https://www.github.com/cianeastwood/qedr`.

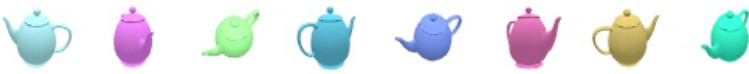

Figure 2: **Data samples.**

is centred on the object, the scene background is removed and additional generative factors (shape and lighting) are held constant. Each generative factor is independently sampled from its respective uniform distribution: azimuth($z_0$) $\sim U[0, 2\pi]$, elevation($z_1$) $\sim U[0, \pi/2]$, red($z_2$) $\sim U[0, 1]$, green($z_3$) $\sim U[0, 1]$, blue($z_4$) $\sim U[0, 1]$. We divide the images into training (160,000), validation (20,000) and test (20,000) sets before removing images which contain particular generative factor combinations to faciliate the evaluation of zeroshot performance (see Appendix B.2). This left 142,927, 17,854 and 17,854 images in the training, validation and test sets respectively.

## 4.2 MODELS

Generative modelling has become one of the leading approaches to unsupervised representation learning, with several recent works imposing additional learning constraints to encourage the model to learn disentangled representations (Desjardins et al., 2012; Reed et al., 2014; Zhu et al., 2014; Cohen & Welling, 2014a; Cheung et al., 2014; Larsen et al., 2015; Makhzani et al., 2015; Chen et al., 2016; Higgins et al., 2017). Of these models, it can be argued that $\beta$-VAE (Higgins et al., 2017) and InfoGAN (Chen et al., 2016) are the most promising due to their scalability and lack of assumptions about the underling factors of variation. Thus, we evaluate the codes learned by these models and compare them to the VAE($\beta = 1$) and PCA.

For fair comparison, we train all models with 10 code variables and use the same network architectures for the VAE, $\beta$-VAE and InfoGAN. More specifically, we use the same residual networks (ResNets, He et al. 2016) for the encoders/discriminator and the decoders/generator (see Table 2), and train InfoGAN with 10 continous 'latent codes'(Chen et al., 2016) and 0 noise variables. We found that these ResNets produced the sharpest images and best visual disentanglement for all generative models, outperforming popular architectures for ($\beta$-)VAE (Larsen et al., 2015; Higgins et al., 2017) and InfoGAN (Kulkarni et al., 2015). We fit $\beta = 6$ and $\lambda = 6$ for $\beta$-VAE and InfoGAN respectively by balancing reconstruction/generation quality and visual disentanglement (see Appendix D), where $\lambda$ is the mutual information coefficient. See Appendix A for further details.

## 4.3 REGRESSORS

**Lasso.** We begin with linear regressors and encourage a *sparse* mapping between $c$ and $z$ with an $\ell_1$ regularisation penalty (lasso regressors). With the inputs and targets normalised to have zero mean and unit variance, the magnitude of the resulting regression weights rank the learnt code variables $c_0, \ldots, c_{D-1}$ in order of *relative importance* to the prediction. That is, they reveal which code variables capture information about a given generative factor. Thus, we define the matrix of relative importances $R$ as $R_{ij} = |W_{ij}|$ for linear regression, where $R_{ij}$ denotes the relative importance of $c_i$ in predicting $z_j$ and $|W_{ij}|$ denotes the magnitude of the weight used to scale $c_i$ in predicting $z_j$. We fit the $\ell_1$ penalty coefficient $\alpha$ on the validation set to achieve the lowest prediction error.

**Random forest.** We use random forest regressors due to their inbuilt ability to determine the relative importance of each feature to a given prediction, thus allowing us to directly specify the matrix of relative importances $R$. Random forests average the predictions and feature importances from each decision tree in the ensemble. The number of times a tree chooses to split on a particular input variable determines its importance to the prediction. Thus, the relative importance of each input variable $c_i$ is given by the number of cases split on $c_i$ over the total number of splits (Breiman et al., 1984). As performance generally improves with the number of trees $n$ in the ensemble, we fix $n = 10$. The remaining parameter, tree depth, is determined on the validation set (lowest prediction error).

| (a) **Lasso** | | | | (b) **Random forest** | | | |
| --- | --- | --- | --- | --- | --- | --- | --- |
| **Code** | **Disent.** | **Compl.** | **Inform.** | **Code** | **Disent.** | **Compl.** | **Inform.** |
| PCA | 0.29 | 0.32 | 0.44 | PCA | 0.50 | 0.52 | 0.27 |
| VAE ($\beta = 1$) | 0.67 | 0.62 | 0.37 | VAE ($\beta = 1$) | 0.86 | 0.75 | 0.09 |
| $\beta$-VAE ($\beta = 6$) | 0.66 | 0.59 | 0.35 | $\beta$-VAE ($\beta = 6$) | 0.90 | 0.76 | 0.10 |
| InfoGAN | 0.75 | 0.72 | 0.23 | InfoGAN | 0.91 | 0.87 | 0.13 |

Table 1: **Average model scores.** 'Inform.' indicates (average) normalised root-mean-square error (NRMSE) in predicting $z$.

## 4.4 Results

**Overview.** Table 1 presents the average disentanglement, completeness and informativeness scores for PCA, the VAE, $\beta$-VAE and InfoGAN for the (a) lasso and (b) random forest regressors. With both regressors, PCA achieves the worst disentanglement, completeness and informativeness scores (highest error in predicting $z$) while the VAE ($\beta = 1$) and $\beta$-VAE ($\beta = 6$) achieve very similar disentanglement, completeness and informativeness scores to each other. While InfoGAN achieves the best disentanglement, completeness and informativeness scores with the lasso regressor, the VAE and $\beta$-VAE achieve similar disentanglement and informativeness scores to it with (the increased capacity of) the random forest regressor.

**Disentanglement.** InfoGAN achieves the highest average disentanglement with both regressors (although $\beta$-VAE closely follows with the random forest regressor). That is, each variable in Info-GAN's code ($c-$InfoGAN) is closest (on average) to capturing a single generative factor, making $c-$InfoGAN the most *disentangled* code (see Appendix B.1 for the full / per-variable results). Figure 3 helps to identify the generative factors captured by a given code variable and thus visualise the disentanglement. For example, comparing $c_0$ across all models in Figure 3 (the first rows), it is clear that $c_0-$VAE, $c_0-\beta$VAE and $c_0-$InfoGAN (almost) solely capture information about $z_0$ while $c_0-$PCA captures information about almost all generative factors.

**Completeness.** InfoGAN also achieves the highest average completeness with both regressors. That is, $c-$InfoGAN is closest (on average) to capturing each generative factor with a single code variable, making $c-$InfoGAN the most *complete* code. In contrast, the low completeness score (overcompleteness) of $c-$PCA reveals that it uses several code variables to capture each generative factor (again, see Appendix B.1 for the full / per-variable results). Figure 3 helps to identify the code variables which capture a given generative factor and thus visualise the completeness. For example, Figure 3 shows that several 'dead' or redundant code variables ($c_5, c_6, c_7$) enable a high degree of completeness in $c-$InfoGAN. In addition, comparing $z_0$ (azimuth) across all models in Figure 3b (the first columns), it is clear that InfoGAN uses three code variables ($c_0, c_1, c_8$) to capture $z_0$ while PCA, VAE, and $\beta$-VAE use significantly more. In particular, Figure 3 shows that $c-$PCA is severely overcomplete in capturing $z_0$, with each of its constituent variables capturing distinct information about $z_0$. With an ideal code, Figure 3 would show a single large square in $K$ rows and each column, indicating a one-to-one mapping between $z$ and $K$ of the dimensions in $c$.

**Informativeness.** With the lasso regressor, $c-$InfoGAN is most predictive of / informative about $z$. That is, $c-$InfoGAN contains the most easily-extractable / explicit information about $z$. This is supported by Figure 5, which plots each $z$ against the corresponding 'most important' code variable(s) (as indicated by the $R$ matrix) and reveals (primarily) linear relationships between $z$ and $c-$InfoGAN. Despite being significantly deeper with many more parameters, $c-$VAE and $c-\beta$VAE are only slightly more predictive of $z$ than $c-$PCA with this linear regressor, indicating that the information about $z$ in $c-$VAE and $c-\beta$VAE is not easily-extractable / explicit (again, this is supported by the relationships depicted in Figure 5).

All model codes better predict $z$ with the random forest regressor, particularly $c-$VAE and $c-\beta$VAE. In fact, $c-$VAE and $c-\beta$VAE are the most predictive of / informative about $z$ with this non-linear regressor, with the increased capacity allowing significantly more information about $z$ to be extracted from these codes. As discussed in section 2, the prediction error with this (non-

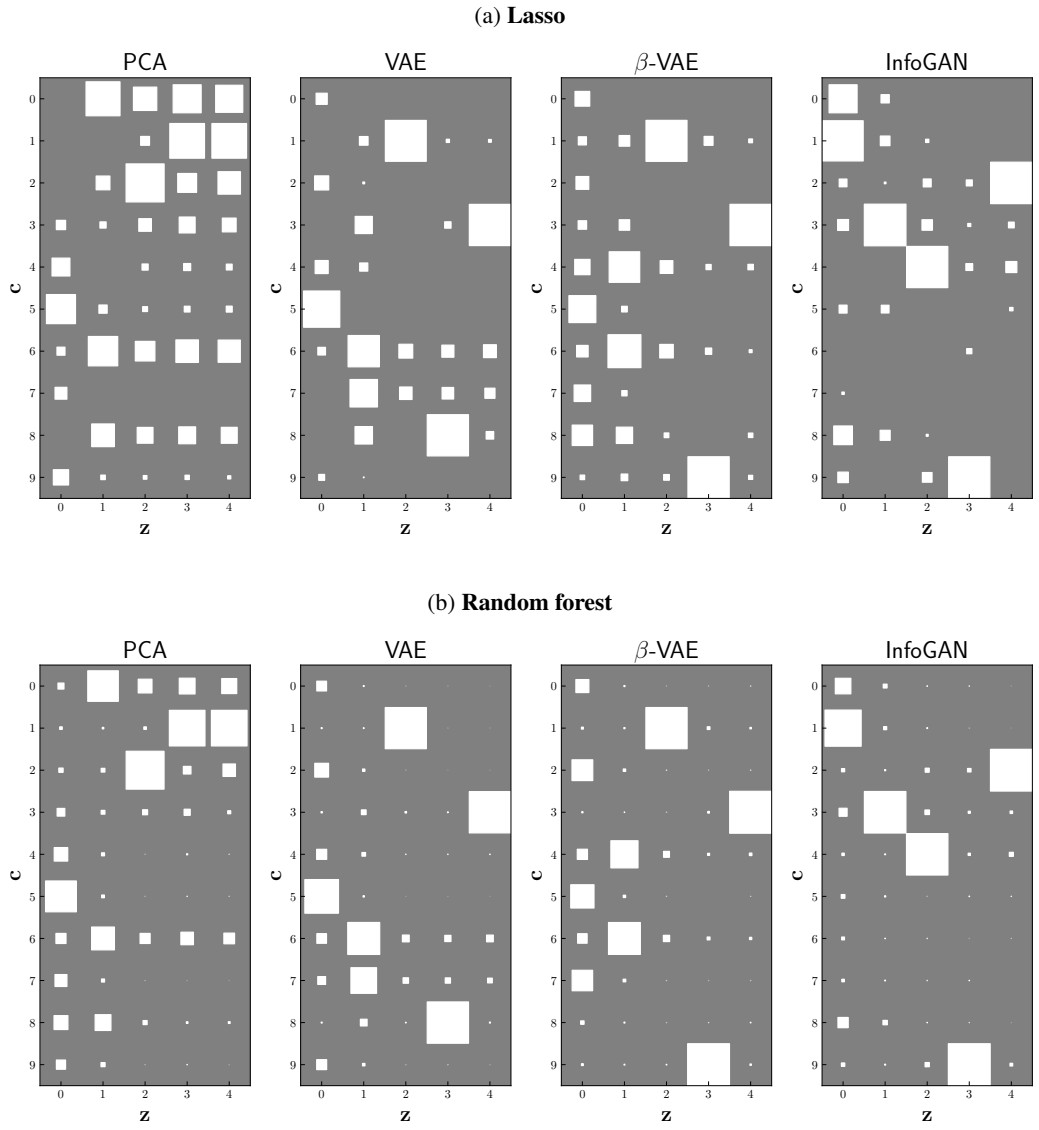

Figure 3: **Visualising $R$**. Square size indicates magnitude, i.e. relative importance. Row $i$ illustrates the importance of $c_i$ to each prediction and thus the disentanglement. Column $j$ illustrates the importance of each code variable for predicting $z_j$ and thus the completeness.

linear) regressor is likely a better quantification of informativeness as it is less dependent on the ability of the model to explicitly represent information about $z$ in $c$.

## 5 CONCLUSION

In this work we have presented a framework for the quantitative evaluation of disentangled representations when the ground-truth latent structure is available. The quality of learnt representations is elucidated through the explicit definition and quantification of three criteria: disentanglement, completeness and informativeness. To illustrate the appropriateness of our framework, we employed it to compare quantitatively the codes learned by PCA, the VAE, $\beta$-VAE and InfoGAN.

While our framework is limited to synthetic datasets where it is possible to recover $z$, reliable disentanglement is far from solved even in this restricted setting. Hence, we believe our framework

and its constituent metrics take a substantial and important step forward in understanding learned representations. We have made the code and dataset publicly available in the hope that this facilitates further model comparisons and eventually the establishment of quantitative benchmarks for disentangled factor learning. While we have focused on image data in this work, future work may explore the applicability of our framework to other types of synthetic data.

## 6 ACKNOWLEDGEMENTS

We would like to thank Pol Moreno and Akash Srivastava for helpful discussions. We would also like to thank Pol for generating the dataset. Finally, we would like to thank the anonymous reviewers for their constructive criticisms which were helpful in refining this paper. The work of CW is supported in part by EPSRC grant EP/N510129/1 to the Alan Turing Institute.

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

## A EXPERIMENTAL SETUP

For all generative models, we use the ResNet architectures shown in Table 2 for the encoder / discriminatior ($D$) / auxilary network ($Q$) and the decoder / generator ($G$). We optimize using Adam (Kingma & Ba, 2014) with a learning rate of 1e-4 and a batch size of 64. For the stable training of InfoGAN, we fix the latent codes' standard deviations to 1 and use the objective of the improved Wasserstein GAN (IWGAN) (Gulrajani et al., 2017), simply appending InfoGAN's approximate mutual information penalty. As in Gulrajani et al. (2017), we use layer normalization (Ba et al., 2016) instead of batch normalization (Ioffe & Szegedy, 2015) in $D$. As in Chen et al. (2016), $Q$ shares all convolutional layers with the discriminator (or 'critic' with WGAN objective) $D$, each adding their own final output layer. As $Q$ parametrises the approximate posterior over continous latent codes $Q(c|x)$, we simply take the mean returned by $Q(x)$ as the code or representation for a given image. Further details on the experimental setup are provided in our open-source implementation.

| Encoder / $D$ / $Q$ | Decoder / $G$ |
|---|---|
| 3×3 64 conv. | FC 4·4·8·64 |
| BN, ReLU, 3×3 64 conv | BN, ReLU, 3×3 512 conv ↑ |
| BN, ReLU, 3×3 128 conv, ↓ | BN, ReLU, 3×3 512 conv |
| BN, ReLU, 3×3 128 conv | BN, ReLU, 3×3 256 conv ↑ |
| BN, ReLU, 3×3 256 conv, ↓ | BN, ReLU, 3×3 256 conv |
| BN, ReLU, 3×3 256 conv | BN, ReLU, 3×3 128 conv ↑ |
| BN, ReLU, 3×3 512 conv, ↓ | BN, ReLU, 3×3 128 conv |
| BN, ReLU, 3×3 512 conv | BN, ReLU, 3×3 64 conv ↑ |
| BN, ReLU, 3×3 512 conv, ↓ | BN, ReLU, 3×3 64 conv |
| FC Output | BN, ReLU, 3×3 3 conv, tanh |

Table 2: **($\beta$-)VAE / InfoGAN architecture.** Each network has 4 residual blocks (all but the first and last rows). The input to each residual block is added to its output (with appropriate downsampling/upsampling to ensure that the dimensions match). Downsampling ($\downarrow$) is performed with mean pooling. $\uparrow$ indicates nearest-neighbour upsampling. When batch normalization (BN) is applied to convolutional layers, per-channel normalization is used.

## B EXTENDED RESULTS

### B.1 FULL TABLE / PER-FACTOR RESULTS

Tables 3 and 4 give the full regression results, i.e. the per-factor disentanglement, completeness and informativeness. As each target is normalised to have a standard deviation of 1, the root-mean-square error (RMSE) in predicting each target is naturally normalised relative to the constant regressor which guesses the expected value of the targets. Hence, we report the NRMSE.

### B.2 ZEROSHOT

Disentangled representations should enable a model to perform zero-shot inference, that is, generalise its knowledge beyond the training distribution by recombining previously-learnt factors (Bengio et al., 2013; Higgins et al., 2017). Thus, we can further evaluate the disentangled representations learned by a given model by quantifying its ability to perform zero-shot inference. We use the ground-truth values of the generative factors to create two different data distributions. More specifically, we isolate all images whose generative factor values lie in a particular range to create a 'gap' in the original dataset. This gap then serves as our zero-shot data containing unseen factor combinations. Informally, the images in this gap can be described as 'red' teapots from 'above'. Formally, the generative factors of these images satisfy the following condition: $z_2 > (z_3 + 0.15)$ and $z_2 > (z_4 + 0.15)$ and $z_1 > \frac{\pi}{4}$. This dataset contained 21,238 images, with (extreme) samples given in Figure 4. Note that zero-shot inference is facilitated by disentangled and informative representations, thus is not a core component of our evaluation, but rather a 'bonus'.

(a) **Disentanglement**

| Code | $c_0$ | $c_1$ | $c_2$ | $c_3$ | $c_4$ | $c_5$ | $c_6$ | $c_7$ | $c_8$ | $c_9$ | W. Avg. |
|---|---|---|---|---|---|---|---|---|---|---|---|
| PCA | 0.16 | 0.50 | 0.31 | 0.09 | 0.45 | 0.60 | 0.11 | 1.00 | 0.17 | 0.51 | 0.29 |
| VAE | 1.00 | 0.85 | 0.95 | 0.68 | 0.63 | 1.00 | 0.30 | 0.37 | 0.66 | 0.95 | 0.67 |
| $\beta$-VAE | 1.00 | 0.64 | 1.00 | 0.76 | 0.39 | 0.89 | 0.49 | 0.81 | 0.45 | 0.80 | 0.66 |
| InfoGAN | 0.83 | 0.85 | 0.76 | 0.66 | 0.78 | 0.43 | 1.00 | 1.00 | 0.64 | 0.74 | 0.75 |

(b) **Completeness**

| Code | $z_0$ | $z_1$ | $z_2$ | $z_3$ | $z_4$ | Avg. |
|---|---|---|---|---|---|---|
| PCA | 0.38 | 0.39 | 0.34 | 0.24 | 0.25 | 0.32 |
| VAE | 0.54 | 0.37 | 0.75 | 0.73 | 0.73 | 0.62 |
| $\beta$-VAE | 0.14 | 0.39 | 0.70 | 0.85 | 0.88 | 0.59 |
| InfoGAN | 0.42 | 0.72 | 0.75 | 0.86 | 0.84 | 0.72 |

(c) **Informativeness**

| Code | $z_0$ | $z_1$ | $z_2$ | $z_3$ | $z_4$ | Avg. |
|---|---|---|---|---|---|---|
| PCA | 0.83 | 0.42 | 0.32 | 0.32 | 0.33 | 0.44 |
| VAE | 0.61 | 0.60 | 0.23 | 0.21 | 0.21 | 0.37 |
| $\beta$-VAE | 0.80 | 0.41 | 0.19 | 0.19 | 0.18 | 0.35 |
| InfoGAN | 0.48 | 0.13 | 0.23 | 0.16 | 0.15 | 0.23 |

Table 3: **Lasso regression results**. **(a)** Disentanglement scores for each code variable. 'W. Avg.' abbreviates weighted average. **(b)** Completeness scores for each generative factor. $z_0, \ldots, z_4$ represent azimuth, elevation, red, green and blue generative factors respectively. **c)** Test set NRMSE.

(a) **Disentanglement**

| Code | $c_0$ | $c_1$ | $c_2$ | $c_3$ | $c_4$ | $c_5$ | $c_6$ | $c_7$ | $c_8$ | $c_9$ | W. Avg. |
|---|---|---|---|---|---|---|---|---|---|---|---|
| PCA | 0.25 | 0.54 | 0.63 | 0.10 | 0.88 | 0.97 | 0.16 | 0.90 | 0.41 | 0.72 | 0.50 |
| VAE | 0.98 | 0.99 | 0.91 | 0.94 | 0.75 | 0.99 | 0.56 | 0.56 | 0.92 | 0.86 | 0.86 |
| $\beta$-VAE | 0.96 | 0.96 | 0.95 | 0.99 | 0.63 | 0.96 | 0.69 | 0.94 | 0.64 | 0.98 | 0.90 |
| InfoGAN | 0.85 | 0.97 | 0.91 | 0.84 | 0.94 | 0.68 | 0.86 | 0.70 | 0.71 | 0.92 | 0.91 |

(b) **Completeness**

| Code | $z_0$ | $z_1$ | $z_2$ | $z_3$ | $z_4$ | Avg. |
|---|---|---|---|---|---|---|
| PCA | 0.31 | 0.51 | 0.67 | 0.56 | 0.56 | 0.52 |
| VAE | 0.44 | 0.61 | 0.90 | 0.91 | 0.91 | 0.75 |
| $\beta$-VAE | 0.28 | 0.67 | 0.90 | 0.96 | 0.97 | 0.76 |
| InfoGAN | 0.59 | 0.94 | 0.91 | 0.96 | 0.95 | 0.87 |

(c) **Informativeness**

| Code | $z_0$ | $z_1$ | $z_2$ | $z_3$ | $z_4$ | Avg. |
|---|---|---|---|---|---|---|
| PCA | 0.36 | 0.23 | 0.20 | 0.28 | 0.28 | 0.27 |
| VAE | 0.14 | 0.09 | 0.09 | 0.06 | 0.06 | 0.09 |
| $\beta$-VAE | 0.18 | 0.07 | 0.08 | 0.09 | 0.08 | 0.10 |
| InfoGAN | 0.25 | 0.07 | 0.14 | 0.09 | 0.10 | 0.13 |

Table 4: **Random forest regression results**. Caption of Table 3 applies.

Table 5 presents the zeroshot results. With the random forest regressor, $c-$VAE and $c-\beta$VAE perform the best with very little increase in prediction error compared to Table 4c, while $c-$InfoGAN predicts the value of unseen factor combinations reasonably well.

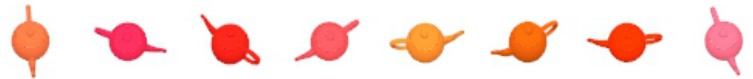

Figure 4: **Zeroshot samples**

|  | (a) **Lasso** |  |  |  |  |  |  | (b) **Random Forest** |  |  |  |  |  |
|---|---|---|---|---|---|---|---|---|---|---|---|---|---|
| **Code** | $z_0$ | $z_1$ | $z_2$ | $z_3$ | $z_4$ | **Avg.** | **Code** | $z_0$ | $z_1$ | $z_2$ | $z_3$ | $z_4$ | **Avg.** |
| PCA | 0.88 | 0.80 | 0.75 | 0.52 | 0.54 | 0.70 | PCA | 0.44 | 0.49 | 0.65 | 0.56 | 0.63 | 0.55 |
| VAE | 0.56 | 1.11 | 0.52 | 0.30 | 0.32 | 0.56 | VAE | 0.13 | 0.13 | 0.34 | 0.08 | 0.07 | 0.15 |
| $\beta$-VAE | 0.81 | 0.79 | 0.32 | 0.27 | 0.25 | 0.49 | $\beta$-VAE | 0.18 | 0.18 | 0.21 | 0.12 | 0.14 | 0.16 |
| InfoGAN | 0.49 | 0.34 | 0.93 | 0.36 | 0.33 | 0.49 | InfoGAN | 0.27 | 0.18 | 0.63 | 0.21 | 0.25 | 0.31 |

Table 5: **Zeroshot performance**. NRMSE in predicting unseen factor combinations.

## C   Z vs. C

Figure 5 plots each generative factor against the corresponding 'most important' code variable(s) (as indicated by $R$) of each model for 5000 randomly-selected samples. As discussed in section 2, models with generic priors cannot be expected to learn the most complete and explicit representation of topologically distinct factors of variation. Thus, for the wrapped azimuth ($z_0$), we plot its value against the 3 most important code variables for each model. Inspecting the relationships depicted in Figure 5, it is clear that the simplest / lowest-order relationship exists between InfoGAN's code variables and the corresponding generative factors. For example, each unwrapped generative factor ($z_1, z_2, z_3, z_4$) is linearly-related to InfoGAN's corresponding code variables, while $c_0$ and $c_8$ resemble scaled sine and cosine functions of the azimuth ($z_0$) and $c_1$ resembles a step function.

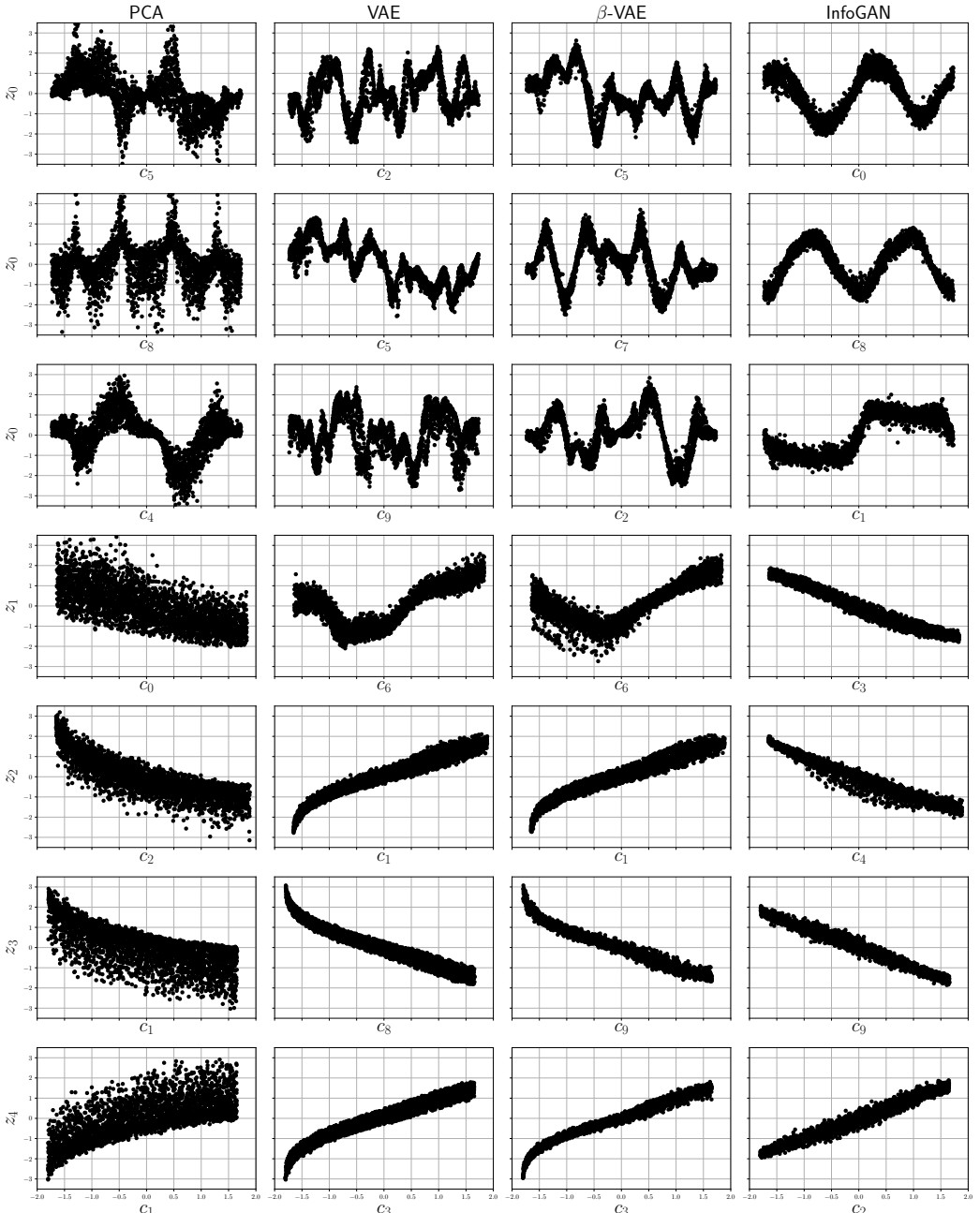

Figure 5: **Generative factors vs. important code variables.**

# D    VISUALLY ASSESSING DISENTANGLEMENT

For each model, we traverse the space of each code variable indepedently to show the effect on generated images and thus visually assess disentanglement. The code variable traversals depicted in Figure 6 for (a) VAE $[-3, 3]$, (b) $\beta$-VAE $[-3, 3]$ and (c) InfoGAN $[-1, 1]$ are ordered according to the generative factor ($z$) which that code best captures in an attempt to align the generated images of all models. There appears to be a high degree of disentanglement in all generative models as each $c_i$ traversal results in a single type of semantic variation.

(a) **VAE**

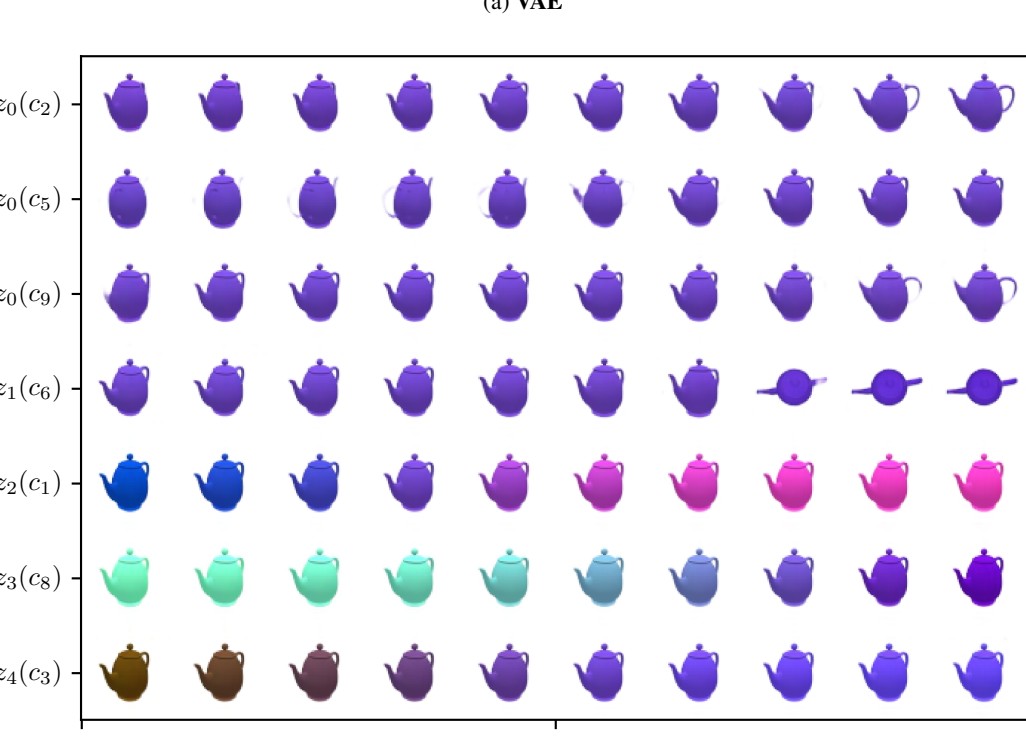

(b) $\beta$-VAE

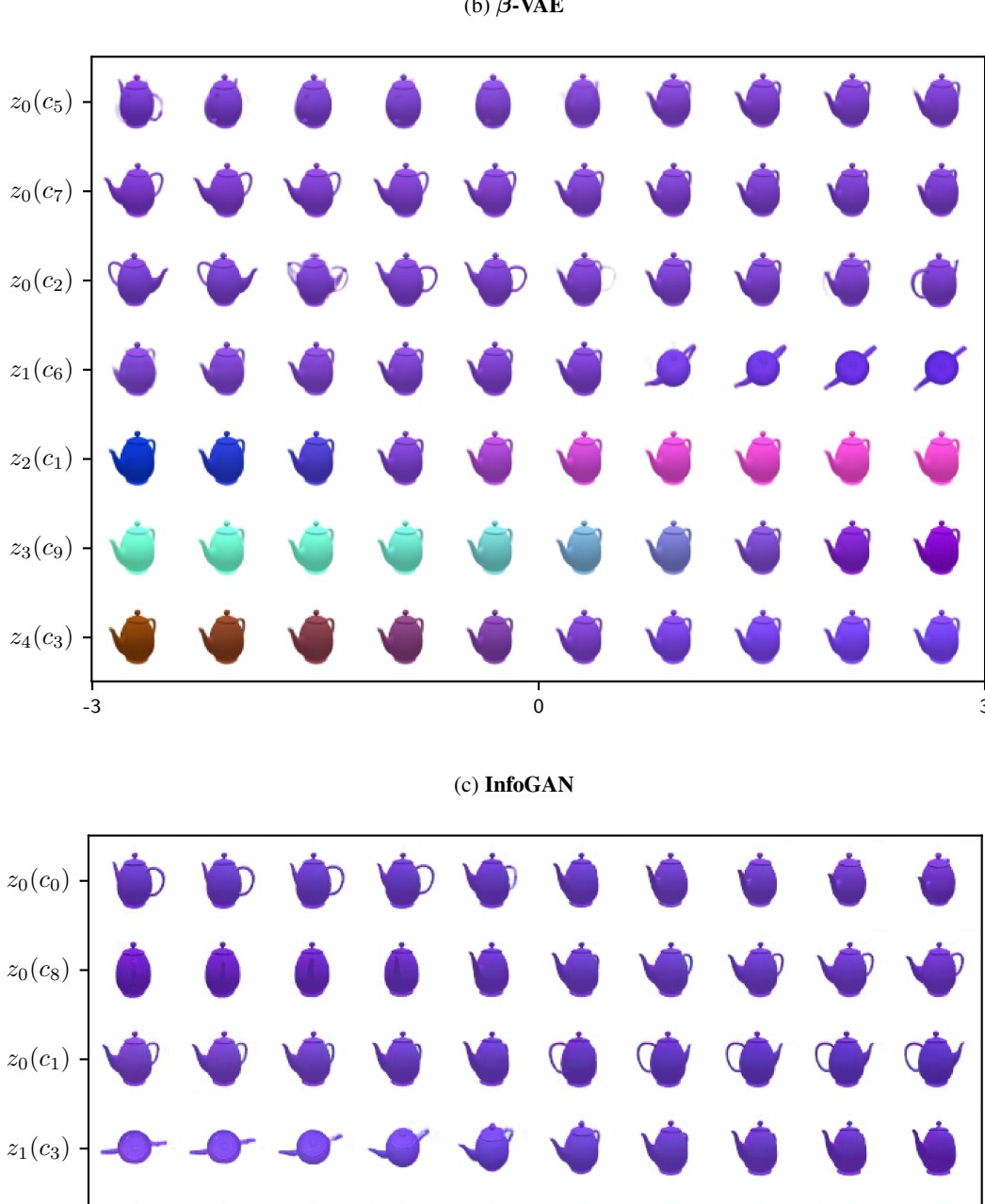

(c) InfoGAN

Figure 6: **Code variable traversals.**

