# OpenReview forum: "A Framework for the Quantitative Evaluation of Disentangled Representations"
_ICLR.cc/2018/Conference — Accept (Poster)_

### Official Review · AnonReviewer3 · 2017-11-27
**This paper has good formal ideas about evaluating disentangling representations, a solid step forward on understanding learned representations. It would benefit from more thoughts on basic theory and more empirical work to be seen as a relevant benchmark going forward.**

**Rating:** 7
**Confidence:** 5

**Review:**

****
I acknowledge the author's comments and improve my score to 7.
****

Summary:
The authors propose an experimental framework and metrics for the quantitative evaluation of disentangling representations.
The basic idea is to use datasets with known factors of variation, z, and measure how well in an information theoretical sense these are recovered by a representation trained on a dataset yielding a latent code c.
The authors propose measures disentanglement, informativeness and completeness to evaluate the latent code c, mostly through learned nonlinear mappings between z and c measuring the statistical relatedness of these variables.
The paper ultimately is light on comprehensive evaluation of popular models on a variety of datasets and as such does not quite yield the insights it could.

Significance:
The proposed methodology is relevant, because disentangling representations are an active field of research and currently are not evaluated in a standardized way.

Clarity:
The paper is lucidly written and very understandable.

Quality:
The authors use formal concepts from information theory to underpin their basic idea of recovering latent factors and have spent a commendable amount of effort on clarifying different aspects on why these three measures are relevant.
A few comments:
1. How do the authors propose to deal with multimodal true latent factors? What if multiple sets of z can generate the same observations and how does the evaluation of disentanglement fairly work if the underlying model cannot be uniquely recovered from the data?
2. Scoring disentanglement against known sources of variation is sensible and studied well here, but how would the authors evaluate or propose to evaluate in datasets with unknown sources of variation?
3. the actual sources of variation are interpretable and explicit measurable quantities here. However, oftentimes a source of variation can be a variable that is hard or impossible to express in a simple vector z (for instance the sentiment of a scene) even when these factors are known. How do the authors propose to move past narrow definitions of factors of variation and handle more complex variables? Arguably, disentangling is a step towards concept learning and concepts might be harder to formalize than the approach taken here where in the experiment the variables are well-behaved and relatively easy to quantify since they relate to image formation physics.
4. For a paper introducing a formal experimental framework and metrics or evaluation I find that the paper is light on experiments and evaluation. I would hope that at the very least a broad range of generative models and some recognition models are used to evaluate here, especially a variational autoencoder, beta-VAE and so on. Furthermore the authors could consider applying their framework to other datasets and offering a benchmark experiment and code for the community to establish this as a means of evaluation to maximize the impact of a paper aimed at reproducibility and good science.

Novelty:
Previous papers like "beta-VAE" (Higgins et al. 2017) and "Bayesian Representation Learning With Oracle Constraints" by Karaletsos et al (ICLR 16) have followed similar experimental protocols inspired by the same underlying idea of recovering known latent factors, but have fallen short of proposing a formal framework like this paper does. It would be good to add a section gathering such attempts at evaluation previously made and trying to unify them under the proposed framework.

---

> ### Author Response · Authors · 2018-01-05
> **Response**
>
> Thank you for your feedback. Please see our response to reviewer 2, which addresses the points made by all three reviewers.

---

### Official Review · AnonReviewer1 · 2017-11-28
**The idea is interesting and, despite the limited experimental setup, the proposed framework can ease future investigations**

**Rating:** 6
**Confidence:** 5

**Review:**

The paper addresses the problem of devising a quantitative benchmark to evaluate the capability of algorithms to disentangle factors of variation in the data.

*Quality*
The problem addressed is surely relevant in general terms. However, the contributed framework did not account for previously proposed metrics (such as equivariance, invariance and equivalence). Within the experimental results, only two methods are considered: although Info-GAN is a reliable competitor, PCA seems a little too basic to compete against. The choice of using noise-free data only is a limiting constraint (in [Chen et al. 2016], Info-GAN is applied to real-world data).
Finally, in order to corroborate the quantitative results, authors should have reported some visual experiments in order to assess whether a change in c_j really correspond to a change in the corresponding factor of variation z_i according to the learnt monomial matrix.

*Clarity*
The explanation of the theoretical framework is not clear. In fact, Figure 1 is straight in identifying disentanglement and completeness as a deviation from an ideal bijective mapping. But, then, the authors missed to clarify how the definitions of D_i and C_j translate this requirement into math.
Also, the criterion of informativeness of Section 2 is split into two sub-criteria in Section 3.3, namely test set NRMSE and Zero-Shot NRMSE: such shift needs to be smoothed and better explained, possibly introducing it in Section 2.

*Originality*
The paper does not allow to judge whether the three proposed criteria are original or not with respect to the previously proposed ones of [Goodfellow et al. 2009, Lenc & Vedaldi 2015, Cohen & Welling 2014, Jayaraman & Grauman 2015].

*Significance*
The significance of the proposed evaluation framework is not fully clear. The initial assumption of considering factors of variations related to graphics-generated data undermines the relevance of the work. Actually, authors only consider synthetic (noise-free) data belonging to one class only, thus not including the factors of variations related to noise and/or different classes.

PROS:
The problem faced by the authors is interesting

CONS:
The criteria of disentanglement, informativeness & completeness are not fully clear as they are presented.
The proposed criteria are not compared with previously proposed ones - equivariance, invariance and equivalence [Goodfellow et al. 2009, Lenc & Vedaldi 2015, Cohen & Welling 2014, Jayaraman & Grauman 2015]. Thus, it is not possible to elicit from the paper to which extent they are novel or how they are related..
The dataset considered is noise-free and considers one class only. Thus, several factors of variation are excluded a priori and this undermines the significance of the analysis.
The experimental evaluation only considers two methods, comparing Info-GAN, a state-of-the-art method, with a very basic PCA.


**FINAL EVALUATION**
The reviewer rates this paper with a weak reject due to the following points.
1) The novel criteria are not compared with existing ones [Goodfellow et al. 2009, Lenc & Vedaldi 2015, Cohen & Welling 2014, Jayaraman & Grauman 2015].
2) There are two flaws in the experimental validation:
	2.1) The number of methods in comparison (InfoGAN and PCA) is limited.
	2.2) A synthetic dataset is only considered.

The reviewer is favorable in rising the rating towards acceptance if points 1 and 2 will be fixed.

**EVALUATION AFTER AUTHORS' REBUTTAL**
The reviewer has read the responses provided by the authors during the rebuttal period. In particular, with respect to the highlighted points 1 and 2, point 1 has been thoroughly answered and the novelty with respect previous work is now clearly stated in the paper. Despite the same level of clarification has not been reached for what concerns point 2, the proposed framework (although still limited in relevance due to the lack of more realistic settings) can be useful for the community as a benchmark to verify the level of disentanglement than newly proposed deep architectures can achieve. Finally, by also taking into account the positive evaluation provided by the fellow reviewers, the rating of the paper has been risen towards acceptance.

---

> ### Author Response · Authors · 2018-01-05
> **Response**
>
> Thank you for your feedback. Please see our response to reviewer 2, which addresses the points made by all three reviewers.

---

### Official Review · AnonReviewer2 · 2017-11-29
**A reasonable if somewhat ad-hoc approach**

**Rating:** 6
**Confidence:** 4

**Review:**

The authors consider the metrics for evaluating disentangled representations. They define three criteria: Disentanglement, Informativeness, and Completeness. They  learning a linear mapping from the latent code to an idealized set of disentangled generative factors, and then define information-theoretic measures based on pseudo-distributions calculated from the relative magnitudes of weights. Experimental evaluation considers a dataset of 200k images of a teapot with varying pose and color.

I think that defining metrics for evaluating the degree of disentanglement in representations is  great problem to look at. Overall, the metrics approached by the authors are reasonable, though the way pseudo-distribution are define in terms of normalized weight magnitudes is seems a little ad hoc to me.

A second limitation of the work is the reliance on a "true" set of disentangled factors. We generally want to learn learning disentangled representations in an unsupervised or semi-supervised manner, which means that we will in general not have access supervision data for the disentangled factors. Could the authors perhaps comment on how well these metrics would work in the semi-supervised case?

Overall, I would say this is somewhat borderline, but I could be convinced to argue for acceptance based on the other reviews and the author response.

Minor Commments:

- Tables 1 and 2 would be easier to unpack if the authors were to list the names of the variables (i.e. azimuth instead of z_0) or at least list what each variable is in the caption.

- It is not entirely clear to me how the proposed metrics, whose definitions all reference magnitudes of weights, generalize to the case of random forests.

---

> ### Author Response · Authors · 2018-01-03
> **Response to all reviewers (part 1)**
>
> We thank the reviewers for their helpful comments, and appreciate the
> view that "defining metrics for evaluating the degree of
> disentanglement in representations is great problem to look at".
>
> Two reviewers raise the issue that our work requires a "true" set of
> generative factors in order to carry out the evaluation. Our response
> is that if it is not possible to quantify disentanglement in this
> situation, it will certainly be much more difficult to quantify it
> when the ground truth is not known, and this must be the first step.
> We have now emphasized in the abstract, introduction and conclusion
> that the method applies when the ground truth generative factors are
> known.
>
> R1:
> > However, the contributed framework did not account for previously proposed
> > metrics (such a equivariance, invariance and equivalence).
> > ...
> > The paper does not allow to judge whether the three proposed criteria
> > are original or not with respect to the previously proposed ones of
> > [Goodfellow et al. 2009, Lenc & Vedaldi 2015, Cohen & Welling 2014,
> > Jayaraman & Grauman 2015].
> > …
> > The novel criteria are not compared with existing ones [Goodfellow et al.
> > 2009, Lenc & Vedaldi 2015, Cohen & Welling 2014, Jayaraman & Grauman
> > 2015].
> and
> R3:
> > Previous papers like "beta-VAE" (Higgins et al. 2017) and "Bayesian
> > Representation Learning With Oracle Constraints" by Karaletsos et al
> > (ICLR 16) have followed similar experimental protocols inspired by the
> > same underlying idea of recovering known latent factors, but have
> > fallen short of proposing a formal framework like this paper does. It
> > would be good to add a section gathering such attempts at evaluation
> > previously made and trying to unify them under the proposed
> > framework.
>
> We have added sec 3 to expand the coverage of related work. The
> relationship to equivariance and invariance is covered in the last
> paragraph of sec 3; note that such properties arise naturally
> from a properly disentangled and informative representation.
>
> We have expanded the comparison to Higgins et al. (2017) and
> Karaletsos et al (2016) in sec 3. We have also added here a paragraph
> on similarities/differences to the work of Yang and Amari (1997) wrt
> the evaluation of ICA, following comments we received on the paper.
>
> R2:
> > Within the experimental results, only two methods are considered:
> > although Info-GAN is a reliable competitor, PCA seems a little too
> > basic to compete against.
> > ...
> > The experimental evaluation only considers two methods, comparing
> > Info-GAN, a state-of-the-art method, with a very basic PCA.
> and
> R3:
> > The paper ultimately is light on comprehensive evaluation of popular models
> > on a variety of datasets and as such does not quite yield the insights it
> > could.
> > ...
> > For a paper introducing a formal experimental framework and metrics or
> > evaluation I find that the paper is light on experiments and evaluation. I
> > would hope that at the very least a broad range of generative models and
> > some recognition models are used to evaluate here, especially a variational
> > autoencoder, beta-VAE and so on.
>
> Our experiments highlight the differences between a baseline (PCA) and a
> state-of-the-art method (InfoGAN). This contrastive
> comparison demonstrates the appropriateness of the framework, with the
> three criteria clearly explaining why InfoGAN's learnt code is superior to PCA's
> and the metric scores quantifying this level of superiority. We will make the
> code and dataset publicly available on acceptance of the paper and hope this
> facilitates further comparisons and eventually the establishment of quantitative
> benchmarks for disentangled factor learning. We note e.g. that the authors of
> the beta-VAE have not published their code, which has made conducting the
> requested experiments more difficult.
>
> R3:
> > Furthermore the authors could consider ... offering a benchmark
> > experiment and code for the community to establish this as a means
> > of evaluation to maximize the impact of a paper aimed at
> > reproducibility and good science.
>
> We will be happy to make the dataset and code publicly available
> on acceptance of the paper, as now mentioned in the conclusion.
>
> R2:
> > though the way pseudo-distribution are define in terms of normalized weight
> > magnitudes is seems a little ad hoc to me.
> > ...
> > It is not entirely clear to me how the proposed metrics, whose
> > definitions all reference magnitudes of weights, generalize to the
> > case of random forests.
>
> We thank the reviewer for this feedback. We have now clarified these points
> by defining the relative importances R_{ij} on p2, and discussing the definition
> of importances for random forests as per Breiman et al (1984) on p3.

---

> ### Author Response · Authors · 2018-01-03
> **Response to all reviewers (part 2)**
>
> R1:
> > The significance of the proposed evaluation framework is not fully clear. The
> > initial assumption of considering factors of variations related to
> > graphics-generated data undermines the relevance of the work.
>
> We have further clarified the significance of the framework in Section
> 1 & 5. The framework is not restricted to graphics-generated data,
> it could also be used e.g. with speech synthesis.
>
> R1:
> > But, then, the authors missed to clarify how the definitions of D_i
> > and C_j translate this requirement into math.
>
> The descriptions of disentanglement and completeness on p 2&3
> make it clear how D_i and C_j quantify the deviation from an
> ideal bijective mapping.
>
> R1:
> > Also, the criterion of informativeness of Section 2 is split into two
> > sub-criteria in Section 3.3, namely test set NRMSE and Zero-Shot
> > NRMSE: such shift needs to be smoothed and better explained, possibly
> > introducing it in Section 2.
>
> We thank the reviewer for pointing this out. We have now clarified (sec 4.1 final
> sentence) that the zero-shot inference task is a "bonus", and not a core
> component of the framework.
>
> R1:
> > The dataset considered is noise-free and considers one class
> > only. Thus, several factors of variation are excluded a priori and
> > this undermines the significance of the analysis.
>
> It would be easy to add noise (e.g. Gaussian) to the output of the
> renderer, but we do not believe that this would have a substantial
> effect on the results. It would be interesting to expand the
> experiments to cover more object classes, but we believe that the
> framework and experiments presented already constitute a substantial
> advance.
>
> R3:
> > 1. How do the authors propose to deal with multimodal true latent
> > factors? What if multiple sets of z can generate the same observations
> > and how does the evaluation of disentanglement fairly work if the
> > underlying model cannot be uniquely recovered from the data?
>
> If multiple sets of z can generate the same observations, then this
> should be reflected in a (multimodal) distribution within the codes
> c. If this were present then it would be propagated via the mapping f
> from c to z into a distribution over z's. Current methods like InfoGAN
> tend to make a unimodal assumption about Q(c|x), but if this were
> multimodal then the above mechanism would work, and one could use the
> obvious log-likelihood criterion log p(z|c) to train the
> regression network (e.g. like a mixture of experts, Jacobs et al
> 1991). Of course the ordinary least squares criterion is just a
> special case of this with a Gaussian noise model for p(z|c).
> We have also added a paragraph at the bottom of p3 concerning the
> rotation-of-factors case, for which the model is not identifiable.
>
> R1:
> > ... in order to corroborate the quantitative results, authors
> > should have reported some visual experiments in order to assess
> > whether a change in c_j really correspond to a change in the
> > corresponding factor of variation z_i according to the learnt monomial
> > matrix.
>
> Please see Figure 6, as per the original submission.
>
> R3:
> > 3. the actual sources of variation are interpretable and explicit
> > measurable quantities here. However, oftentimes a source of variation
> > can be a variable that is hard or impossible to express in a simple
> > vector z (for instance the sentiment of a scene) even when these
> > factors are known. How do the authors propose to move past narrow
> > definitions of factors of variation and handle more complex variables?
> > Arguably, disentangling is a step towards concept learning and
> > concepts might be harder to formalize than the approach taken here
> > where in the experiment the variables are well-behaved and relatively
> > easy to quantify since they relate to image formation physics.
>
> It is vital to be able to quantify disentangling wrt what R3 calls a
> "simple" vector z. The contribution of the paper is to do this. There
> may well be more complex sources of latent structure, such as the
> inter-relationship of different objects in a scene. In our view
> this can likely be handled by an appropriate hierarchical
> model with a vector of z's at the highest level, but this is
> an issue for future work.
>
> R2:
> > Tables 1 and 2 would be easier to unpack if the authors were to list
> > the names of the variables (i.e. azimuth instead of z_0) or at least
> > list what each variable is in the caption.
>
> fixed.

---

### Decision · Program_Chairs · 2018-01-29
**ICLR 2018 Conference Acceptance Decision**

**Decision:**

Accept (Poster)

**Comment:**

The paper proposes evaluation metrics for quantifying the quality of disentangled representations. There is consensus among reviewers that the paper makes a useful contribution towards this end. Authors have addressed most of reviewers' concerns in their response.